# From Intestinal Epithelial Homeostasis to Colorectal Cancer: Autophagy Regulation in Cellular Stress

**DOI:** 10.3390/antiox11071308

**Published:** 2022-06-30

**Authors:** Qiuluo Liu, Yan Chen, Li Zhou, Haining Chen, Zongguang Zhou

**Affiliations:** 1Department of Gastrointestinal Surgery, West China Hospital, Sichuan University, No. 17, Block 3, Southern Renmin Road, Chengdu 610041, China; liuqiuluo@stu.scu.edu.cn; 2State Key Laboratory of Biotherapy and Cancer Center, West China Hospital, Sichuan University, and Collaborative Innovation Center for Biotherapy, Chengdu 610041, China; yanchen0524@scu.edu.cn (Y.C.); 2015224060079@stu.scu.edu.cn (L.Z.)

**Keywords:** autophagy, intestinal homeostasis, colorectal cancer, stress response

## Abstract

The intestinal epithelium is continuously exposed to abundant stress stimuli, which relies on an evolutionarily conserved process, autophagy, to maintain its homeostasis by degrading and recycling unwanted and damaged intracellular substances. Otherwise, disruption of this balance will result in the development of a wide range of disorders, including colorectal cancer (CRC). Dysregulated autophagy is implicated in the regulation of cellular responses to stress during the development, progression, and treatment of CRC. However, experimental investigations addressing the impact of autophagy in different phases of CRC have generated conflicting results, showing that autophagy is context-dependently related to CRC. Thus, both inhibition and activation of autophagy have been proposed as therapeutic strategies against CRC. Here, we will discuss the multifaceted role of autophagy in intestinal homeostasis and CRC, which may provide insights for future research directions.

## 1. Introduction

The mucosal surface of the gastrointestinal tract continuously encounters abundant stimuli originating from both endogenous and exogenous sources, including metabolic alterations, a variety of bacterial species, chemical irritants, and agents that produce oxidative stress. Autophagy, a stress-responsive process, is tightly linked to the maintenance of intestinal cellular homeostasis [1,2] (Figure 1). Under conditions of physiological stress, cells in the intestinal mucosa frequently accumulate unwanted and damaged intracellular substances. In this case, autophagy can be triggered to transport them to the lysosomes for degradation and recycling [3]. Intestinal epithelial cells (IECs) and intestinal stem cells rely on this mechanism to ensure their survival, as it helps maintain protein and organelle quality by selectively degrading and recycling aggregates of impaired or unnecessary proteins, mitochondria, peroxisomes, and endoplasmic reticulum (known as selective autophagy) [4,5,6]. Moreover, autophagic degradation of the intestinal tight junction proteins governs the intensity of the intestinal barrier. Apart from this, autophagy plays a central role in the host–microbiota interactions, where it eliminates potential pathogens and forms an integral component of anti-infectious immunity [2].

In contrast, defective autophagy predisposes normal IECs to undergo malignant transformation. Although the exact etiological mechanisms underlying CRC remain multifactorial and largely unknown, it is well established that both genetic predisposition and environmental factors contribute to its initiation and development. The genetic basis underpinning sporadic CRCs is well defined by theories such as the adenoma–carcinoma sequence model, suggesting that CRC is driven by sequential genetic and epigenetic mutations, arising from normal epithelial cells to dysplastic adenomas and, ultimately, carcinomas [7,8,9,10]. Various genetic events are required during the malignant transformation, involving mutations of *Adenomatous polyposis coli (APC)*, *KRAS*, and *P53* [7,8,10]. It is essential to perceive that across stages of CRC tumorigenesis, alteration of autophagy-related genes plays a significant role. A large genome-wide association study identified genetic variants of transcription factor EB (*TFEB*), a positive regulator of the autophagic pathway that promotes the expression of autophagy genes [11,12], as a novel risk factor associated with CRC susceptibility [13]. Mutation of another autophagy regulator, UV-radiation-resistance associated gene (*UVRAG*), which activates the Beclin1-PI3KC3 complex, also underpins the genetic basis of CRC tumorigenesis [14]. Similarly, genetic alterations involved in the endocytosis-autophagy network were frequently observed in *KRAS*-wild-type CRC [15].

Given that it generally takes years to decades for adenomas to transform into carcinomas, the mutated precursor cells constantly endure endogenous and exogenous stress [16,17]. A clear role has emerged for autophagy in CRC cells, where it exerts diverse effects on cellular adaptation to tumor microenvironmental cues and therapeutic stress, which ultimately results in cell survival, death, or growth inhibition [18,19]. First, highly proliferative CRC cells tend to have a limited supply of nutrients. In the context of nutrient deprivation, autophagy is triggered to provide energy sources and metabolites to sustain metabolism and tumor growth [20,21,22]. Moreover, insufficient and irregular neovascularization of rapidly proliferating CRC cells causes a hypoxic microenvironment, where autophagy is harnessed to eliminate protein aggregates and damaged endoplasmic reticulum (ER) and mitochondria [4,5]. This contributes to the prevention of the overproduction of reactive oxygen species (ROS) and reduction of oxidative and ER stress, thereby preserving genomic integrity [23]. In addition, intestinal microorganisms with oncogenic properties continuously cause an abnormal microenvironment that profoundly affects the initiation and progression of sporadic CRC [24]. The involvement of autophagy in the interaction of microbiota and CRC is complicated and differs in a temporal manner [1,2]. Finally, abnormal autophagy is activated in response to treatment and confers resistance to therapeutic challenges. In this context, autophagy protects CRC cells from drug-induced apoptosis and induces them into a slow-cycling, drug-tolerant state.

In this review, we focus on the regulatory roles of autophagy in the maintenance of intestinal homeostasis. Meanwhile, we discuss how dysregulation of this conserved process orchestrates different stress factors in a context-dependent manner in distinct stages of CRC development and progression and under therapeutic pressure, with the aim of providing a perspective for future research.

## 2. Autophagy Maintains Intestinal Epithelial Homeostasis under Physiological Stress

The intestinal mucosa is constantly exposed to alimentary and bacterial antigens as well as mechanical stress, which relies on an intact intestinal barrier and healthy gut microbiota to maintain intestinal homeostasis that would otherwise cause infection, inflammation, and cellular damage [1,35,36,37]. Regulation of autophagy plays a key role in the ability of the gut epithelium to cope with cell stress, as elucidated by lines of evidence from experimental and clinical studies (Figure 2).

The IECs constitute the first line of defense, which includes the formation of the physical barrier as well as the integration of regulatory mechanisms. As the intestinal epithelium is one of the most vigorously regenerative tissues in adults, its turnover serves as a crucial mechanism for the protective effect provided by the mucosal barrier, which is achieved through a balance of cell apoptosis and proliferation in crypts [38]. Mouse models with ATG14 or Rb1cc1/Fip200 deleted in the intestinal epithelium exhibited extensive intestinal villous atrophy, suggesting that autophagy is protective against cell death during homeostasis in the intestinal epithelium [39]. Mechanistically, these autophagy-related proteins defended intestinal epithelial cells from TNF (tumor necrosis factor)-triggered apoptosis [39]. Moreover, intestinal homeostasis is maintained by leucine-rich repeat-containing G protein-coupled receptor 5-positive intestinal stem cells (LGR5^+^ ISCs) for constant tissue regeneration. Notably, autophagy has been demonstrated to play a cytoprotective role in the LGR5^+^ ISCs against toxic and infectious injuries. During irradiation damage, muramyl dipeptide (MDP), a microbiota-derived product, can be recognized by NOD2 in LGR5^+^ ISCs, thereby promoting cell survival by mediating the clearance of ROS. This reduction of ROS was achieved via mitophagy induction coordinated by NOD2 and ATG16L1, which eliminate damaged mitochondria in ISCs and therefore enhance epithelial repair [40]. In addition, autophagy contributes to LGR5^+^ ISCs maintenance under conditions of irradiation and chemotherapy. Following these stresses, ATG7-dependent DNA damage repair was stimulated, facilitating ISCs survival. Activation of autophagy on fasting showed a protective effect on LGR5^+^ ISCs against oxaliplatin and doxorubicin-induced DNA damage and cell death [41].

The balance of host–microbiota interactions has profound impacts on the host’s intestinal health. Notably, the crucial role of autophagy lies in maintaining intestinal microbiota homeostasis, and dysfunctional autophagy is known to cause gut microbial dysbiosis [42]. As previously demonstrated in the mouse model with conditional inactivation of Atg5 in IECs, blockade of the autophagic flux led to a remarkable alteration and reduced the diversity of gut microbiota [43]. The altered colonization pattern involved decreased abundances of anti-inflammatory microorganisms and enrichment of proinflammatory bacterial groups, many of which are believed to be associated with inflammatory bowel disease (IBD) and colorectal cancer (CRC) [43].

The proposed mechanisms by which autophagy modulates the balance of bacterial flora include direct degradation of harmful bacteria and regulation of the antibacterial immune response. Under physiological conditions, all intestinal bacteria are coated with complement protein C3 [44]. Following the invasion of potentially pathogenic microorganisms into the intestinal mucosa, C3 on the bacterial surfaces can be targeted by host cytosol ATG16L1, thereby activating the autophagy system [45]. Apart from this, it has been reported that MyD88, the canonical adaptor for inflammatory signaling pathways, was also required during the process of autophagy induction [46]. It is worth mentioning that autophagy in IECs can affect the expression and secretion of antimicrobial peptides (AMPs) to restrict bacterial dissemination [47]. Interestingly, several mouse models in which different autophagy genes are deleted in IECs (including *ATG16L1*, *ATG4B* and *LC3B)* showed an enhanced response to microbiota-induced type I interferon (IFN-I) signaling [48]. This spontaneous activation of IFN-I in IECs conferred protection against the pathogen *Citrobacter rodentium* and chemical injury via C-C motif chemokine receptor 2 (CCR2)-dependent monocyte recruitment, fortifying the intestinal barrier in response to both infectious and non-infectious stress [48]. Although autophagy was demonstrated to have an adverse function in antimicrobial activity and tissue repair, as evidenced by this study, the immunomodulatory properties of IFN-I signaling may be far more nuanced under different circumstances, such as autoimmune diseases and tumor immunity [48,49].

Epithelial cells in the intestinal tract attach via tight junctions (Tjs) including claudin, occludin, etc. TJ modulation is closely linked to intestinal permeability, and autophagy has been implicated in enhancing intestinal barrier function via TJ regulation [1]. By mediating the lysosomal-dependent degradation of claudin-2, a pore-forming protein, starvation-induced autophagy reduced intestinal permeability of ions and small molecules in IECs [50]. Further mechanistic investigation revealed that autophagy-triggered claudin-2 degradation was dependent on clathrin-mediated endocytosis, where claudin-2 directly binds to adaptor related protein complex 2 subunit mu 1 (AP2M1), and an increased claudin-2-AP2M1-LC3 association was observed [51]. In contrast, proinflammatory cytokine tumor necrosis factor alpha (TNF-α) weakened the intestinal barrier. This was mediated by the inhibitory effect of TNF-α on autophagy, which resulted in elevated claudin-2 expression and impaired epithelial tight junction [52]. Another TJ-associated protein, occludin, is also tightly regulated by autophagy in IECs. Notably, beclin 1 interacted with occludin on the cell membrane, leading to the endocytosis of occludin and, subsequently, defective TJ barrier function. While this process was autophagy independent, autophagy activation was shown to counteract the effect of Beclin 1 and restore the endothelial barrier [53].

Overall, these findings revealed that autophagy is required for the maintenance of intestinal homeostasis, but its beneficial or deleterious nature can vary depending on the setting.

## 3. Autophagy Coordinates Cellular Adaptation to Stress in the Progression of CRC

### 3.1. Autophagy Enables Adaptation to Metabolic Alteration-Induced Stress

After oncogenic transformation, the established tumor is highly proliferative and metabolically active, requiring large amounts of energy and metabolic precursors. Unlike other normal cells, cancer cells constitutively utilize glycolysis to sustain oncogenic metabolism [54]. In such a situation, autophagy serves as a mechanism of survival [22]. Given that a key feature of autophagy has been suggested to supply substrates to fuel metabolism, it is constantly active under nutrient-competent conditions to enhance tumor growth [55]. Cancer cells utilize it as an alternative source of nearly all aspects of metabolic fuel and reduce oxidative stress, creating “autophagy addiction” [55,56,57]. Otherwise, autophagy-deficient tumor cells suffer from metabolic vulnerabilities and energy crises in the stressed microenvironment. Indeed, in the transformed IECs, not the adjacent normal IECs, autophagy is indispensable for cell metabolism [42]. Deletion of *ATG7* in intestinal adenoma blocked its progression to malignancy via p53-induced growth arrest and AMPK-dependent downregulation of glycolytic genes [42], consistent with the theory that cancer cells exhibit a particular addiction to autophagy [58,59].

Starvation-induced reduction of nutrient inputs, such as glucose and amino acids, leads to decreased intermediate metabolites of various metabolic pathways; for example, the tricarboxylic acid (TCA) cycle [60]. This will ultimately lower the ratio of adenosine triphosphate (ATP) to adenosine diphosphate (ADP) and adenosine monophosphate (AMP), energy stress that can be sensed by AMPK [61]. Once activated, AMPK suppresses ATP-consuming pathways and upregulates energy-generating processes, including autophagy [62]. In contrast, another major autophagy modulator, mTOR, is sensitive to the abundance of amino acids and is activated by available nutrients. In addition, other regulators including ATF4, SIRT1, and TFEB govern the transcription of autophagy-related genes, in response to nutrient availability and reduction status [62]. Together, there is an intricate regulatory network that integrates autophagy with response to metabolic cues.

Interestingly, the metabolic reliance on autophagy of tumors may be dependent on their mutational status. Prior studies have revealed different degrees of autophagy addiction across multiple carcinoma types, most notably in Ras-driven tumors, including lung cancer, pancreatic cancer, prostate cancer, and CRC [58,63,64,65]. *KRAS* is mutated in approximately 40% of CRC patients and is associated with poor prognosis and therapy resistance [66]. For Ras-transformed CRC cells, enhanced glucose metabolism is required for their high rates of proliferation in starvation. Autophagy has been shown to facilitate glycolytic flux in *H-RAS^V12^* cells [63], likely due to its potential to degrade macromolecules and provide metabolic substrates. Interestingly, autophagy is able to protect mitochondrial function in *H-RAS^V12^* and *K-RAS^V12^* cells. This was achieved through the supply of substrates for mitochondrial metabolism, presumably TCA-cycle metabolites, via conversion of pyruvate and fatty acids into acetyl-CoA [64]. Defective autophagy in *H-RAS ^V12^* and *K-RAS^V12^* models also impaired mitochondrial respiration, causing reduced energy levels and increased oxidative stress [64]. However, while lack of autophagy in a mouse model of *KRAS*-driven lung cancer resulted in impaired fatty acid oxidation, it was absent in the *BRAF*-driven mouse model [67]. This raises the question as to the influence of genotype on the metabolic role of autophagy, which is broadly unknown in CRC.

Apart from glucose metabolism, autophagy is closely associated with fatty acid metabolism. Fatty acid β-oxidation in mitochondria produces acetyl CoA, thereby fueling the TCA cycle [68]. Indeed, CRC patient-derived adipocytes were shown to favor the survival of CRC cells under the condition of nutrient deprivation. Mechanistically, the adipocytes secreted free fatty acids, which in turn are absorbed and utilized by colon cancer cells by inducing autophagy and mitochondrial fatty acid β-oxidation via AMPK activation [69]. In addition to cell-autonomous autophagy, host autophagy has a metabolic role in the antitumor immunity of CRC [70]. In activated T_reg_ cells, autophagy was functionally stimulated and negatively regulated mTORC1-dependent glycolytic metabolism, thus promoting their metabolic homeostasis and immunosuppressive function [71]. Together, these studies suggested the metabolic vulnerabilities mediated by autophagy and provided opportunities for therapeutic intervention in CRC (Figure 3).

### 3.2. Autophagy Enables Adaptation to Hypoxia-Induced Stress

Due to rapid proliferation, excessive oxygen consumption, and abnormal microvasculature, the tumor mass of CRC is constantly exposed to reduced oxygen levels. Hypoxia is one of the major hallmarks of the CRC microenvironment, and autophagy is elicited to enable tumor cells to thrive in this situation. Hypoxia-inducible factor (HIF-1), the major transcription factor complex in response to hypoxic conditions, can induce autophagy through upregulation of autophagy-related genes, crosstalk with the mTOR signaling, and production of reactive oxygen species (ROS) [72,73,74]. Notably, under hypoxia stress, functional mitochondria play an indispensable role in ROS generation and subsequent HIF-1 stabilization.

The association of autophagy dysfunction with CRC initiation is evident in prior studies, in which essential autophagy genes, including Atg7 [42], Atg16l1 [75], and UVRAG [14], were edited in mice. In these cases, the arisen neoplasms would accumulate large amounts of autophagic cargo, most obviously damaged mitochondria [76]. Mitochondria is responsible for the adaptation of cells to a variety of stressors, and autophagy functions to eliminate defective mitochondria, a process known as “mitophagy” [77]. Indeed, during the onset of CRC, enhanced mitophagy in IECs was demonstrated to cause lysosomal membrane permeabilization via an iron(II)-dependent mechanism. In turn, the elevated lysosomal permeability led to release of proteases and subsequent antigen presentation, thereby activating CD8^+^ T cells and antitumor immunity [78]. Excess ROS is another type of stress generated by abnormal cellular metabolism, hypoxia, and proteotoxic stress during intestinal tumorigenesis [6]. It was shown that autophagy in IECs was essential for counteracting ROS to enhance barrier integrity, and therefore attenuated the development of CRC [79].

Nevertheless, although autophagy prevents tumor formation at the early stage of intestinal carcinogenesis, it is not the case once the malignant transformation is established. Tumor-initiating cells (TICs), a cell subpopulation endowed with unlimited self-renewal and enhanced tumor-formation capacities, are known to greatly favor CRC initiation [80]. Under hypoxic conditions, autophagy promoted the self-renewal of TICs and their tumorigenic potential [81]. Conversely, autophagy suppressed the growth of the more differentiated counterpart cells [81]. As another example, Atg7-deficiency in IECs attenuated intestinal tumorigenesis in Apc(+/−) mice by the regulation of microbiome-mediated antitumor responses [42]. In samples taken from CRC patients, upregulation of Beclin 1 was related to HIF-1α overexpression, which further correlated with higher histological grade, disease stage, and poor prognosis [82]. Regarding the molecular mechanism, under normoxic conditions, Bcl-xL and Bcl-2 interacted with beclin-1, thereby inhibiting autophagy. In contrast, under hypoxic conditions, HIF-1α promoted the expression of proapoptotic genes BNIP3 and BNIP3L, which are associated with Bcl-xL and Bcl-2 to release beclin-1, thus triggering prosurvival autophagy in CRC cells [72]. In addition, HIF-1α upregulated miR-210, which suppressed Bcl-2 and induced autophagy to reduce the radiosensitivity of CRC [83]. Hypoxia also causes the accumulation of misfolded or unfolded proteins, leading to the unfolded protein response (UPR), and extended UPR signaling promotes cellular apoptosis. In CRC cells, hypoxia elicited UPR and the downstream key factor eukaryotic translation initiation factor 2 alpha kinase 3 (EIF2AK3). Subsequently, EIF2AK3 upregulated transcription factors ATF4 and CHOP to enhance the expression of LC3 and ATG5, thereby triggering cytoprotective autophagy [84]. Moreover, a recent study also demonstrated a sequential activation of AMPK, HIF-1α, HIF-2α, and JNK that accounted for the autophagy induction in CRC cells exposed to low oxygen levels [85]. Thus, these studies highlighted the distinct roles of autophagy in coordinating hypoxia stress response at different stages of colorectal development (Figure 4).

### 3.3. Autophagy Enables Adaptation to Oncogenic Microorganism-Induced Stress

As discussed above, accumulating evidence has supported the involvement of gut microorganisms in intestinal homeostasis and the etiology of sporadic CRC [2,86]. Bacterium pattern is different in CRC patients compared to healthy individuals. Through sequencing studies of the intestinal microbiota, the contribution of certain bacteria, including Fusobacterium nucleatum, Escherichia coli, and Bacteroides fragilis in CRC has been well established [87,88]. These infectious agents trigger DNA damage in host genetics by producing genotoxins, generating carcinogenic metabolites, regulating host cell signaling pathways, and shaping the cancer immune landscape in CRC [89,90,91,92,93,94].

A prime example of the role of microbiota-mediated autophagy in CRC is Fusobacterium nucleatum. Fusobacterium nucleatum, a Gram-negative anaerobe, is frequently present in the oral cavity and is commonly involved in dental plaques and periodontal disease [95]. Of note, Fusobacterium nucleatum was found in approximately 30% of CRC tissues in patients [96], and its abundance was positively associated with lymph node metastasis [97] and worse prognosis [98]. Interestingly, Fusobacterium nucleatum was enriched in CRC tissues from patients who relapsed after chemotherapy [24]. Although 5-fluorouracil (5-FU) in combination with platinum-based chemotherapy has been the first-line therapy for CRC patients [99], most patients develop chemoresistance during treatment and relapse after the initial response [100]. Mechanistic investigations revealed that infection with Fusobacterium nucleatum activated the innate immune response via TLR4 and MYD88-dependent signaling, which resulted in downregulation of miR-4802 and miR-18a*. Subsequently, reduction of these microRNAs attenuated their target on the 3′UTR regions of ULK1 and ATG7 genes, thus alleviating the silencing of autophagy. Eventually, activated autophagy gave rise to chemoresistance to oxaliplatin and 5-fu by protecting CRC cells from drug-induced apoptosis [24]. This has highlighted the prominent role of Fusobacterium nucleatum in coordinating a network of immune responses and autophagy to govern chemoresistance in CRC. Similarly, this network has also been implicated in CRC metastasis. Upon infection, Fusobacterium nucleatum induced the expression of CARD3 in CRC cells, an essential kinase involved in innate and adaptive immune signaling. Upregulation of CARD3 then enhanced autophagic flux, thereby promoting the formation of liver and lung metastases in mouse models [101]. Nevertheless, the specific mechanism by which CARD3 regulates autophagy has been elusive and warrants further investigation.

Another mucosa-associated bacterium, Escherichia coli, is likely to exert an oncogenic phenotype in CRC through crosstalk with autophagy in a time-dependent manner. Activated autophagy can protect against CRC initiation in response to bacterial-induced stress. During the early stage of CRC initiation, increased epithelial autophagy eliminated the intracellular Colibactin-producing Escherichia coli to ameliorate malignant transformation in ApcMin/+ mice [75]. Colibactin-producing *E coli* (CoPEC), a colonic mucosa-associated E coli frequently detected in CRC patients, are able to promote CRC development by inducing genomic instability and inflammation [93,102,103]. There was evidence that autophagy-mediated elimination of Colibactin-producing Escherichia coli (CoPEC) limited the carcinogenesis process in ApcMin/+ mice by stimulating bacteria-induced DNA damage repair via RAD51 and reducing the secretion of inflammatory cytokines IL 6 and IL 8 [75]. Following this, at the mid phase of CRC development, the invasive *E. coli* that successfully colonized the colonic epithelium blocked autophagy to avoid clearance, achieving persistent infection. The repression of autophagy by *E. coli*, in turn, led to increased generation of ROS and epithelial hyperproliferation. However, as the dysplasia tissue progressed, autophagy was upregulated to eradicate the pathogen, suggesting an *E. coli* -independent tumor growth in the late stage of CRC development [104]. In line with this, the only time window when antibiotic intervention exhibited a tumor-suppressive effect was in the middle stages of tumor development [104]. Hence, the interplay between gut microbes and autophagy changes over time in the course of CRC development and therefore awaits future studies (Figure 5).

## 4. Autophagy Modulates Response to Therapeutic Stress in CRC

Although the above studies have highlighted the different roles of autophagy in coordinating environmental cues with CRC tumorigenesis and development, extensive laboratory evidence supported the stimulation of autophagy under therapeutic stress in CRC [105]. Given that numerous stress-sensing signaling pathways that elicit autophagy are utilized by CRC treatment approaches, many of these drugs have been revealed to induce cytoprotective autophagy [106]. In addition to surgery, patients with CRC are treated with combination regimens that involve chemotherapy, radiation therapy, and targeted therapy, tailored to specific pathologic staging and genetic status. Exposure to these therapeutic approaches can trigger autophagy that can enable tumor survival via DNA damage response, ER stress response, mTOR and AMPK signaling, and other stress-activated signaling pathways [105]. Indeed, cytoprotective autophagy has been seen as a crucial mechanism underpinning therapeutic resistance in CRC [107]. Moreover, as demonstrated by several preclinical studies and clinical trials, combining autophagy inhibitors with standard conventional therapies can improve the drug response of CRC [107]. Here, we focus on the roles of autophagy under different therapeutic stresses and the mechanism by which it mediates drug resistance in CRC (Figure 6).

### 4.1. Autophagy Regulates Cellular Response to Chemotherapy

The commonly used chemotherapeutics for treating CRC include fluorouracil (5-FU), oxaliplatin, and irinotecan, alone or in combination [66]. FU, an analog of uracil, mainly suppresses thymidylate synthase, which prevents the generation of thymidine needed for DNA synthesis, thereby inhibiting the proliferation of CRC cells [108]. In CRC models in vitro and in vivo, autophagy activation has been observed upon 5-FU treatment and protected cells from 5-FU-induced apoptosis [109]. The underlying mechanisms involved the upregulation of Bcl-xL, a key crosstalk factor between autophagy and apoptosis, and activation of the P53-AMPK-mTOR pathway [109]. Abnormal activity of metabolic enzymes also contributes to autophagy-mediated chemoresistance. ABHD5, a lipolytic factor situated in the lysosome, binds to PDIA5 to attenuate its inhibitory effect on ribonuclease RNASET2. In turn, RNASET2 regulates RNA degradation in autophagolysosomes, producing oligonucleotides, including uracil [110]. Treatment with 5-FU triggered metabolic reprogramming in CRC cells, and the expression of ABHD5 enhanced autophagic uracil yield, thus conferring 5-FU resistance due to decreased intake of 5-FU as external uracil [111]. Another report observed the induction of autophagy as a key mechanism of irinotecan resistance in TP53-defective CRC cells through the MAPK14/p38α pathway [112]. In addition, autophagy elicited by extracellular cytokine IL-6 protected CRC cells against the cytotoxic effects of 5-FU and oxaliplatin via the JAK2/BECN1 signaling axis [113]. Similar findings were made in microsatellite instability (MSI) CRCs, where mutation of a key autophagy regulator, UVRAG, led to a significant reduction in functional autophagy and became more responsive to 5-FU, oxaliplatin, and irinotecan [114].

The extensive laboratory studies above supported that autophagy engages in a complex interplay with apoptosis under therapeutic stress. Interestingly, apart from apoptosis, autophagy serves as a key mechanism for maintaining cell survival in a drug-tolerant persister (DTP) state to survive the stressful environment caused by chemotherapy [115,116]. In this context, CRC cells reversibly transition into a largely quiescent or slow-growing state, and after withdrawal of treatment, they exit the DTP state and regain the ability of growth and proliferation [117,118]. Mechanistically, this is achieved by employing an evolutionarily conserved program, diapause, which is adopted by hundreds of mammalian species that can suspend embryonic development under unfavorable environmental conditions [119]. Remarkably, as revealed by analyses of expression signatures of the diapause-like DTP in CRC models, this phenotype was maintained via downregulation of the mTOR pathway and upregulation of the autophagy program [120]. Therefore, combination therapy of chemotherapy and autophagy inhibitors represented an innovative therapeutic strategy to disrupt the survival mechanism and prevent cancer relapse [120].

### 4.2. Autophagy Regulates Cellular Response to Targeted Therapies

Aberrant activation or upregulation of oncogenes including EGFR, KRAS, NRAS, and BRAF are frequently present in CRC [121]. Biologics targeting EGFR, such as cetuximab and panitumumab, are often incorporated into the chemotherapy regimens based on the mutation status of individual patients [122,123,124,125]. Anti-EGFR monoclonal antibodies act by blocking access of ligands to the binding domain of EGFR and promoting its internalization and degradation. The interplay between EGFR and autophagy involves RAS PI3K-AKT-mTOR pathways, which serve as the downstream signaling of EGFR as well as the key regulatory network of autophagy [126]. Given the common mechanisms shared by internalized EGFR and autophagy, it is not surprising that activation of autophagy was demonstrated to underlie the acquired resistance of anti-EGFR therapies. Indeed, it has been reported that treatment with the EGFR antibody cetuximab can elicit autophagy in CRC cells and protect them from therapy-induced apoptosis [127,128]. Mechanistic investigations revealed that cetuximab suppressed the expression of HIF-1α and subsequently Bcl-2, which attenuated the inhibitory effect of Bcl-2 on beclin 1 and enhanced the formation of the beclin 1/hVps34 complex, thus activating autophagy [127]. Moreover, cetuximab downregulated miR-216b, which can impair the translation of Beclin-1 through binding to 3′-UTR of its mRNA, thereby inducing cytoprotective autophagy [128]. Therefore, these studies suggested the potential of autophagy inhibitors to sensitize CRC to anti-EGFR monoclonal antibodies.

BRAF-V600E mutation activates the MEK/ERK pathway, conferring a poor prognosis in CRC patients [129]. Targeted combination therapy with BRAF inhibitor encorafenib plus EGFR inhibitor cetuximab has been shown to extend overall survival and approved for second-line therapy [130]. Intriguingly, targeting MEK/ERK pathway using MEK inhibitor trametinib induced prosurvival autophagy by activating the LKB1/AMPK/ULK1 axis in KRAS-mutated pancreatic ductal adenocarcinoma [131]. This was similarly relevant to CRC, since combination therapy of trametinib with autophagy inhibitor chloroquine demonstrated significant antitumor effects in patient-derived xenografts (PDX) of BRAF-mutated CRC [131]. Another monoclonal antibody, bevacizumab, which targets VEGF and interrupts tumor angiogenesis, has been extensively used in CRC [132]. In mouse xenografts of CRC, bevacizumab elicited autophagy and blockade of autophagy with chloroquine displayed synergistic antiproliferative effects against tumor [133]. Photodynamic therapy (PDT), in which photosensitizers are irradiated and excited by light, leads to ROS generation and accumulation, and eventually cell death [134]. This novel technique has become a complement to traditional cancer treatment. Notably, it has been reported that PDT can activate autophagy in CRC, and pharmacological autophagy inhibitors enhanced therapeutic sensitivity to PDT [135]. Together, autophagy serves as a key survival mechanism in response to chemotherapies, targeted therapies, and PDT against CRC; therefore, autophagy inhibition may be an effective therapeutic strategy in CRC.

### 4.3. Autophagy Regulates Cellular Response to Immunotherapy

While autophagy was convincingly shown to be hijacked by cancer cells to resist therapeutic challenges, the consensus that combining autophagy inhibitors with chemotherapy should be regarded as a general therapeutic strategy has been challenged. It is important to perceive that conventional chemotherapies exert anticancer effects not only through a direct cytotoxic mechanism, but also partly owing to the re-stimulation of antitumor immune function [136]. Interestingly, evidence has indicated that autophagy has a major role in immunological control in response to immunogenic chemotherapy in CRC [137,138]. In the context of anticancer chemotherapy exposure, autophagy-competent CRC favored ATP secretion from malignant cells, thereby enhancing the recruitment of dendritic cells and T lymphocytes [137]. Moreover, similar findings were revealed in melanoma, where chemotherapy and radiotherapy-induced autophagy has been shown to augment the sensitivity of tumor cells to lysis by cytotoxic T cells [138,139]. Together, these lines of evidence highlighted that suppression of autophagy might, at least in part, result in a reduction in immunogenicity of cancer cells, and hence defective immune response and relapsed disease.

On theoretical grounds, this detrimental side effect exerted by autophagy inhibition that blunts the antitumor immunity in CRC may be circumvented via combined administration with an immune checkpoint inhibitor [105,140]. Indeed, it has been shown that blocking PIK3C3/VPS34 in combination with anti-PD-1/PD-L1 immunotherapy exhibited promising efficacy in CRC [141,142]. However, in this study, autophagy inhibition achieved by targeting PIK3C3/VPS34 promoted the attraction of cytotoxic immune cells via STAT1/IRF7-dependent production of CCL5 and CXCL10 [141,142]. Along similar lines, experimental studies addressing the impact of autophagy on cancer immune landscape have yielded a wealth of controversial results across various cancer types. In mouse models of melanoma and breast cancer, the levels of T cell infiltration and T cell responses remained unchanged upon autophagy inhibition [143], whereas in other studies, loss of autophagy was believed to facilitate recruitment of antitumor immune effector cells to the tumor bed [144,145,146,147]. The extrapolation can be made that targeting different autophagy proteins may elicit different impacts on cancer immune response and presumably involves autophagy-independent mechanisms. Hence, there is still a lack of knowledge regarding the interaction between autophagy and antitumor immunity.

## 5. Clinical Implications and Future Perspectives

Mounting evidence suggested a prominent role of autophagy in the development of cancer, especially in those organs that are constantly challenged by environmental stressors, such as the large intestine [148]. The idea that autophagy serves as a survival mechanism for tumor cells has provided the logical rationale for autophagy inhibition as a therapeutic strategy in CRC [149]. Indeed, autophagy inhibitors, notably chloroquine (CQ) or hydroxychloroquine (HCQ), have been widely adopted in combination with traditional chemotherapy/radiotherapy in clinical trials of multiple tumor types. Other specific inhibitors are also in development and need further investigation in preclinical and clinical trials [150]. Although the safety of these drugs has been demonstrated, the efficacy of autophagy inhibition has varied widely between patients with different types of tumors and at different stages [148]. These reported clinical outcomes, which are not always encouraging, exemplify the underlying limitations of the clinical applications of autophagy inhibition.

It is critical to note that autophagy has multifaceted and opposing roles in the world of oncology. First, it also plays a cytotoxic role under certain circumstances, which is related to its regulation of apoptosis by the degradation of different proapoptotic or antiapoptotic factors. As such, autophagy inhibition is a bad idea since it would protect malignant cells from undergoing programmed cell death. Moreover, in the context of tumor initiation, growth, and therapeutic pressure of CRC, autophagy functions in a context-dependent manner. For different cell types along the course of the adenoma-carcinoma sequence, including normal IECs, hyperproliferative IECs, adenoma cells, and carcinoma cells, autophagy exerts opposing effects in the presence of distinct microenvironmental conditions. For example, autophagic defects predispose normal cells to malignant transformation, whereas tumor cells can exploit autophagy to thrive under the hostile microenvironment and survive anticancer therapy. In the meantime, while accumulating studies support that autophagy operates in a cell-intrinsic fashion, it also has a cell-extrinsic function. A prime example of this is its relevance in immunological control, where autophagy is responsible for the immunostimulatory signal-sending (notably, ATP) and effector immune cell recruitment [151].Thus, based on these observations, autophagy inhibition may be counterproductive in cancer therapy. To address this dilemma, evaluation with appropriate biomarkers of the status of autophagy, that is, prosurvival or prodeath, whether tumorigenic or tumor-suppressive, may aid in selecting patients who will benefit from autophagy inhibition or induction therapy.

Another issue about the clinical implication of autophagy manipulation is drug specificity. Currently, most pharmacological modulators of autophagy do not selectively target autophagy. Various inhibitors that regulate the different steps of autophagy, including those targeting mTORC1, ULK1, Beclin1, and so on, also interfere with other oncogenic signaling cascades. The ubiquitous effects of autophagy on normal tissues may also limit the clinical utility of autophagy regulators, given that deficiency of autophagy can result in neurodegeneration, lysosomal storage diseases, and other organ dysfunction [152]. Thus, with the increasing understanding of the non-autophagic role of autophagy-related proteins, as well as the potential toxicity of global autophagy modulation on non-transformed tissues, specific regulation of autophagy-related functions local to tumor lesions is required to prevent adverse effects.

Overall, it is impossible to achieve long-term remission and cure through a single-agent treatment in cancer; therefore, combination therapy utilizing multiple means holds great potential for optimal management of CRC [105]. Thus, further explorations that shed additional light on the pleiotropic mechanisms of autophagic machinery more accurately will be critical to help enhance the effectiveness of current CRC therapy.

## Figures and Tables

**Figure 1 antioxidants-11-01308-f001:**
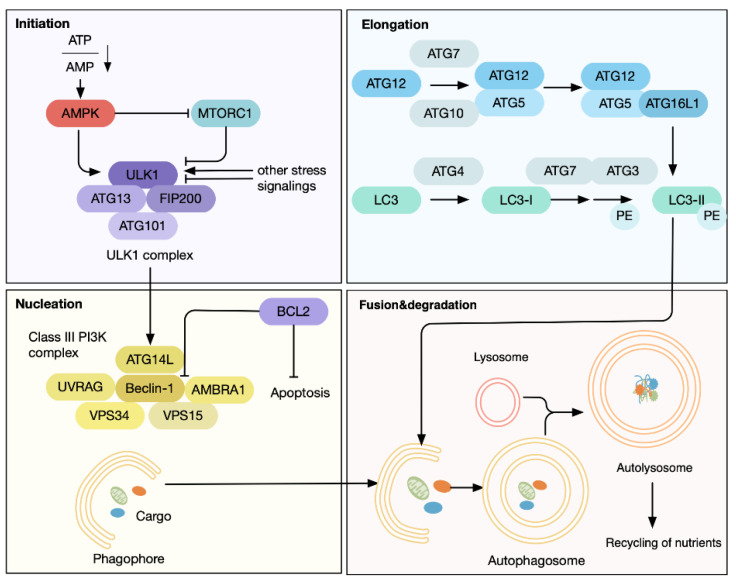
Functional mechanisms of autophagy. The most-studied form of autophagy is macroautophagy, a multistage and dynamic process. Autophagy is negatively regulated by the cell growth promoter rapamycin complex 1 (mTORC1), while the sensor of energy deprivation AMP-activated kinase (AMPK) activates autophagy. The canonical autophagy consists of four sequential stages: initiation, phagophore formation, phagophore elongation, and autophagosome–lysosome fusion [25]. During the initiation phase, the UNC-like autophagy-activating kinase 1 (ULK1) complex serves as a bridge between the upstream mTOR and AMPK and downstream autophagosome formation [26,27]. The complex is composed of ATG13, focal adhesion kinase-family-interacting protein of 200 kDa (FIP200), ATG101, and ULK1, in which ULK1 is the core protein with serine/threonine kinase activity [28]. After being stimulated by nutrient deficiency and stress-related pathways, phosphorylated ULK1 subsequently leads to membrane nucleation, which requires activation of class III phosphatidylinositol-3 kinase complex I (PI3KC3-CI). Formed by beclin-1, vacuolar protein sorting 34 (VPS34), autophagy-related protein 14- like protein (ATG14L), p150, and nuclear receptor-binding factor 2 (NRBF2), this multiprotein complex can be activated through ULK-dependent phosphorylation [29]. The nucleation of the isolation membrane, known as the phagophore, further expands with the support of PI3KC3-CI [30]. At the phagophore assembly site, the complex produces PI3P, favoring the recruitment of the effector proteins (such as WIPI/II), thus resulting in the phagophore elongation [31]. This phase is further promoted by the ubiquitin-like conjugation system, involving the E1 ligase, ATG7, the E2 ligase, ATG3, and the E3 ligase complex, ATG12/ATG5/ATG16L [32]. Through ATG4-dependent proteolytic cleavage, followed by the action of the conjugation system, microtubule-associated proteins 1A/1B light chain 3 (LC3) can be transformed to lipidated LC3 (LC3-II), which is instrumental for elongation and closure of the phagophore [33]. Meanwhile, LC3-II physically links to substrates that contain the LC3-interacting region (LIR) motif, thereby targeting them for degradation. Once phagophores are closed, the ensuing autophagosomes fuse with lysosomes to form autolysosomes; within them, the delivered contents are degraded and recycled [34].

**Figure 2 antioxidants-11-01308-f002:**
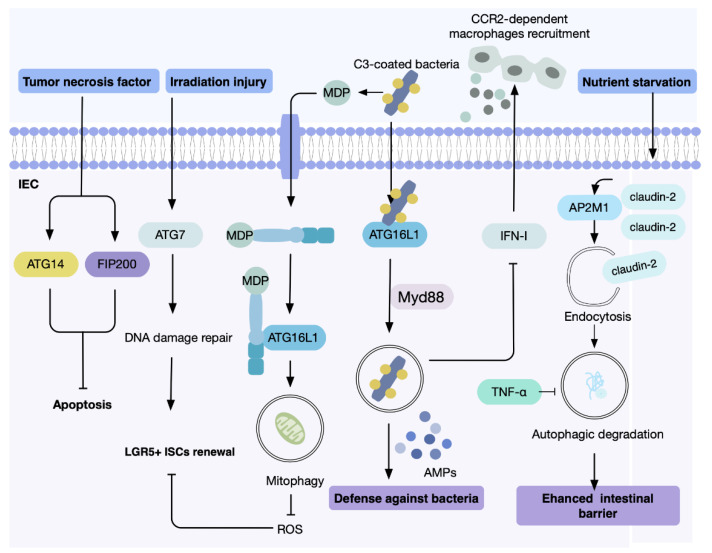
Role of autophagy in intestinal homeostasis maintenance Multiple roles of autophagy in intestinal homeostasis are shown, including regulating the survival of intestinal epithelial cells and intestinal stem cells, the host–microbiota interactions, and the intestinal tight junctions.

**Figure 3 antioxidants-11-01308-f003:**
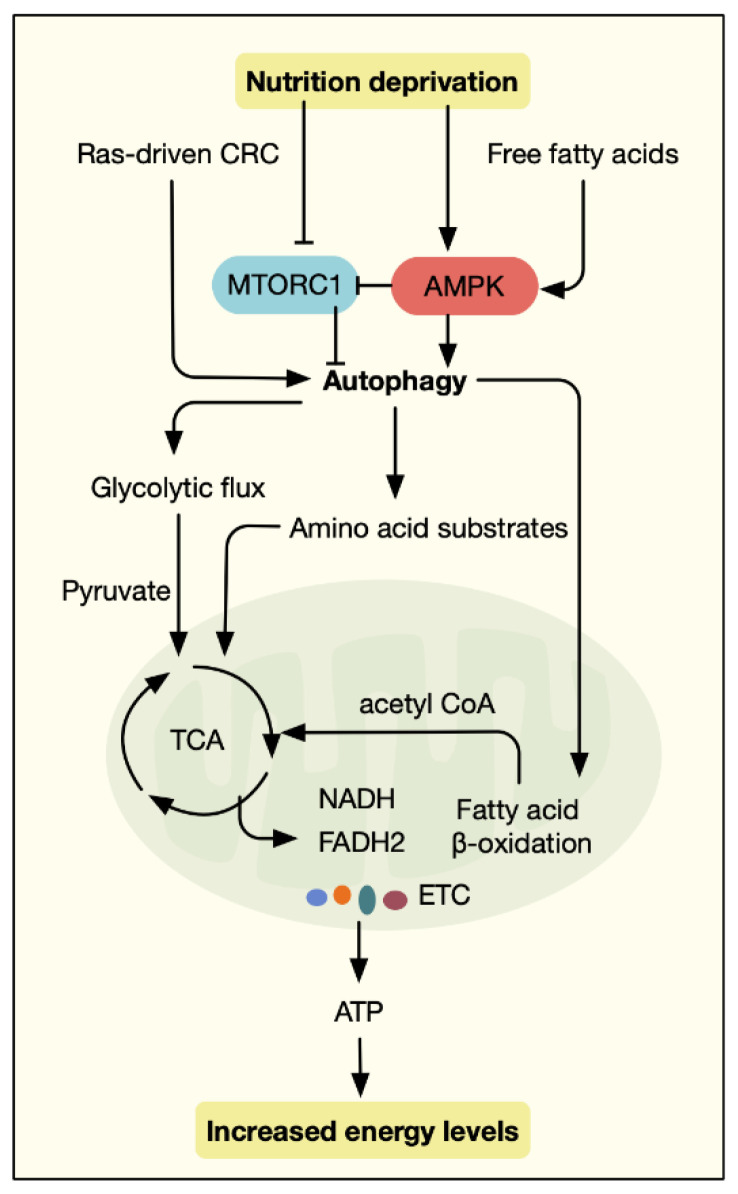
The crosstalk between autophagy and metabolic reprogramming CRC cells is highly proliferative and metabolically active, requiring large amounts of energy and metabolic precursors. In this context, autophagy serves as an alternative source of nearly all aspects of metabolic fuel. By coordinating glycolysis, fatty acid β-oxidation, and tricarboxylic acid cycle, autophagy is intimately connected to the metabolic reprogramming of CRC.

**Figure 4 antioxidants-11-01308-f004:**
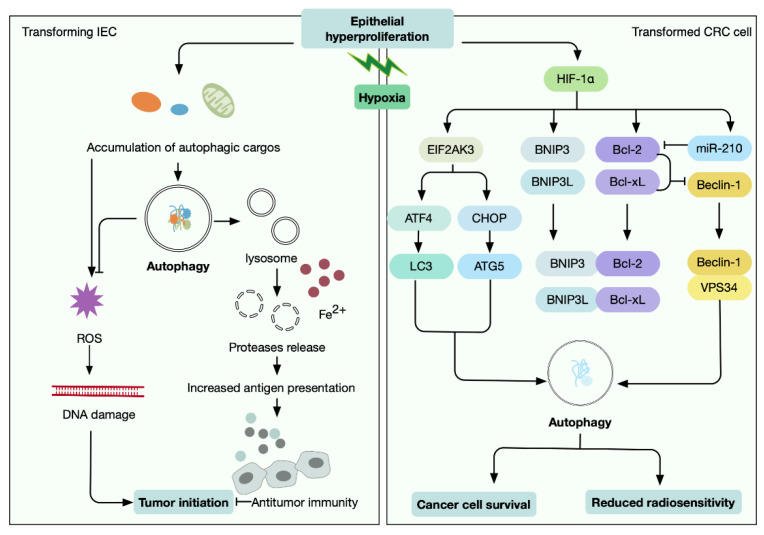
Autophagy coordinates cellular adaptation to hypoxia. Epithelial hyperproliferation results in a reduced level of oxygen, and dysregulated autophagy is involved in response to hypoxia. In transforming IECs, autophagy prevents cancer initiation via the elimination of hypoxia-induced accumulation of damaged cellular substances; while in transformed CRC cells, it promotes cancer cell survival by orchestrating multiple stress response pathways.

**Figure 5 antioxidants-11-01308-f005:**
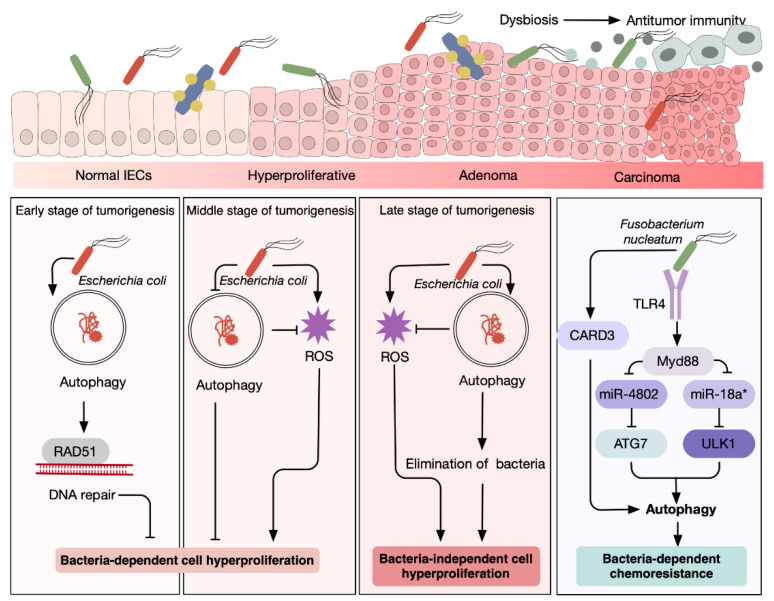
Distinct roles of autophagy in microbiota-induced stress. Sporadic CRC is driven by sequential genetic and epigenetic mutations, and environmental factors, arising from normal epithelial cells to dysplastic adenomas and, ultimately, carcinomas. During this process, a link between CRC tumorigenesis, infection with certain bacteria, and autophagy has been established. Time-dependent interactions between autophagy and intestinal bacteria are shown.

**Figure 6 antioxidants-11-01308-f006:**
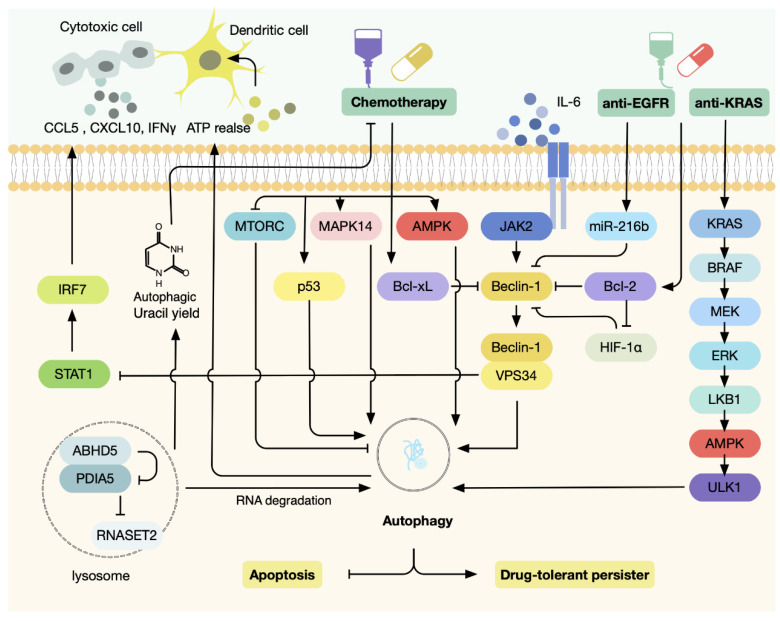
Autophagy under therapeutic stress in CRC. Autophagy activation has been observed during chemotherapies, targeted therapies, and PDT against CRC. In most cases, autophagy serves as a survival mechanism by protecting cells from apoptosis or maintaining cell survival in a DTP state; therefore, autophagy inhibition may be an effective therapeutic strategy in CRC. Paradoxically, autophagy is indispensable in the immune response to chemotherapy in CRC; hence, suppression of autophagy may result in a reduction of immunogenicity of cancer cells and impair antitumor immune immunity. Therefore, whether autophagy inhibitors should be combined with conventional therapies warrants further investigation.

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
