# Peer review of "From Intestinal Epithelial Homeostasis to Colorectal Cancer: Autophagy Regulation in Cellular Stress"

_antioxidants, 2022, doi:10.3390/antiox11071308_

Round 1
Reviewer 1 Report
The article presented by Qiu-Luo Liu and collaborates, entitled “From intestinal epithelial homeostasis to colorectal cancer: autophagy regulation in cellular stress”, is a review that aimed to know the regulatory roles of autophagy in the maintenance of intestinal homeostasis. Also, the authors discuss how dysregulation of the autophagy organizes different stress factors in a context-dependent manner in distinct stages of colorectal cancer development and progression and under therapeutic pressure. The objective of the work is great and pretentious but the authors carry it out with ease. The bibliography used is current and pertinent, it would be appreciated that the authors indicate the keywords used in the search, as well as the databases used, inclusion criteria, etc. to give a more solid scientific basis to the review. To carry out a correct review, the authors must carry out an adequate, broad and sufficient search of scientific documents, and not settle for a few materials (sometimes out of date and out of context) to support their research. That is why I consider it important to include this information in the review, even if it is not a systematic review.
The great contribution of the review are the wonderful figures, which summarize the findings in a concrete, eloquent and clear way. The work is divided into 3 sections, with different subsections, all supported with figures.
Author Response
For Reviewer #1:
The article presented by Qiu-Luo Liu and collaborates, entitled “From intestinal epithelial homeostasis to colorectal cancer: autophagy regulation in cellular stress”, is a review that aimed to know the regulatory roles of autophagy in the maintenance of intestinal homeostasis. Also, the authors discuss how dysregulation of the autophagy organizes different stress factors in a context-dependent manner in distinct stages of colorectal cancer development and progression and under therapeutic pressure. The objective of the work is great and pretentious but the authors carry it out with ease. The bibliography used is current and pertinent, it would be appreciated that the authors indicate the keywords used in the search, as well as the databases used, inclusion criteria, etc. to give a more solid scientific basis to the review. To carry out a correct review, the authors must carry out an adequate, broad and sufficient search of scientific documents, and not settle for a few materials (sometimes out of date and out of context) to support their research. That is why I consider it important to include this information in the review, even if it is not a systematic review.
The great contribution of the review are the wonderful figures, which summarize the findings in a concrete, eloquent and clear way. The work is divided into 3 sections, with different subsections, all supported with figures.
Reply: Thank you for your useful comments. We conducted a systematic literature search in the PubMed database, without language restrictions, to identify relevant studies up to 8th May 2022. The following search strategy was used: (((("Colorectal Neoplasms"[MeSH Terms]) OR ("Intestine, Large"[Mesh])) OR ("Intestinal Mucosa"[Mesh])) AND ("Autophagy"[Mesh])) AND (((((("Gastrointestinal Microbiome"[Mesh]) OR ("Energy Metabolism"[Mesh])) OR ("Hypoxia"[Mesh])) OR ("Drug Therapy"[Mesh])) OR ("Molecular Targeted Therapy"[Mesh])) OR ("Immunotherapy"[Mesh])). Cross-referencing was conducted by manually searching bibliographies of the included articles for additional eligible studies. Articles of interest were those studies that contain classic or groundbreaking discoveries, or elucidate molecular mechanisms in detail, or have far-reaching clinical implications. Two reviewers (Q.L.L. and Y.C.) used these criteria to independently select the eligible studies.
Reviewer 2 Report
This is a literature review concerning intestinal epithelial homeostasis and colorectal cancer.
In my opinion the article is very interesting and the topic promising.
I suggest adding a section / paragraph on clinical implications and future perspectives that can clarify and underline the aim of the study
Moderate English-language editing
Please consider the following articles to be included in the introduction and/or discussion
Fusobacterium nucleatum Promotes Chemoresistance to Colorectal Cancer by Modulating Autophagy. Cell. 2017 Jul 27;170(3):548-563.e16. doi: 10.1016/j.cell.2017.07.008
Mast Cells, microRNAs and Others: The Role of Translational Research on Colorectal Cancer in the Forthcoming Era of Precision Medicine. J Clin Med. 2020 Sep 3;9(9):2852. doi: 10.3390/jcm9092852
The Role of non-coding RNAs in colorectal cancer, with a focus on its autophagy. Pharmacol Ther. 2021 Oct;226:107868. doi: 10.1016/j.pharmthera.2021.107868
Author Response
For Reviewer 2
This is a literature review concerning intestinal epithelial homeostasis and colorectal cancer.
In my opinion the article is very interesting and the topic promising.
I suggest adding a section / paragraph on clinical implications and future perspectives that can clarify and underline the aim of the study
Reply: We appreciated the reviewer’s valuable comments and suggestions. According to your valuable suggestions, we have discussed clinical implications and future perspectives in the revised manuscript [line 556-606].
Moderate English-language editing
Reply: Thank you for your useful comments. Our current manuscript has been polished, and we hope the new revision is now satisfactory.
Please consider the following articles to be included in the introduction and/or discussion
Fusobacterium nucleatum Promotes Chemoresistance to Colorectal Cancer by Modulating Autophagy. Cell. 2017 Jul 27;170(3):548-563.e16. doi: 10.1016/j.cell.2017.07.008
Mast Cells, microRNAs and Others: The Role of Translational Research on Colorectal Cancer in the Forthcoming Era of Precision Medicine. J Clin Med. 2020 Sep 3;9(9):2852. doi: 10.3390/jcm9092852
The Role of non-coding RNAs in colorectal cancer, with a focus on its autophagy. Pharmacol Ther. 2021 Oct;226:107868. doi: 10.1016/j.pharmthera.2021.107868
Reply: Thank you for your valuable comments. We totally agree that these articles should be included in this review. These important references have been added to the manuscript [ref 17, 19, and 24].
Reviewer 3 Report
This is an excellent and erudite overview of the involvement of autophagy in the integrity/homeostasis of the GI tract as well as in the development and therapy of colorectal cancer. The only recommendation from this reviewer is that the authors could more definitively address the limitations to autophagy inhibition as a therapeutic strategy in colon cancer ( this is very well done with regards to the immune system) in that most of the clinical trials addressing this question have not been encouraging in their outcomes.
Author Response
For Reviewer 3
This is an excellent and erudite overview of the involvement of autophagy in the integrity/homeostasis of the GI tract as well as in the development and therapy of colorectal cancer. The only recommendation from this reviewer is that the authors could more definitively address the limitations to autophagy inhibition as a therapeutic strategy in colon cancer ( this is very well done with regards to the immune system) in that most of the clinical trials addressing this question have not been encouraging in their outcomes.
Reply: We appreciate this suggestion and are sorry for the limited description of this topic. A brief discussion on the limitations of autophagy inhibition as a therapeutic strategy in colorectal cancer has been added in the revised manuscript [line 556-606].
Round 2
Reviewer 1 Report
The autors have answered the questions
Reviewer 3 Report
With the modifications that the authors have made, this work is now acceptable for publication.